# Investigation of the inhibitory potential of secondary metabolites isolated from *Fernandoa adenophylla* against Beta-glucuronidase *via* molecular docking and molecular dynamics simulation studies

Abdur Rauf[ID][1]*, Rahaf Ajaj[2]*, Zuneera Akram[3], Majid Khan[4], Abdul Wadood[5], Maryam Zulfat[5], Zafar Ali Shah[6], Abdulhakeem S. Alamri[ID][7,8], Walaa F. Alsanie[7,8], Majid Alhomrani[7,8], Humaira Hussain[4], Dorota Formanowicz[ID][9]

1 Department of Chemistry, University of Swabi, Swabi, Anbar, Khyber Pakhtunkhwa (K.P.), Pakistan, 2 Department of Environmental and Public Health, College of Health Sciences, Abu Dhabi University, Abu Dhabi, United Arab Emirates, 3 Department of Pharmacology, Faculty of Pharmaceutical Sciences, University of Karachi, Karachi, Pakistan, 4 Department of Biochemistry, Abbottabad University of Science & Technology, Abbottabad, Pakistan, 5 Department of Biochemistry, Abdul Wali Khan University Mardan, Mardan, KPK, Pakistan; 6 Department of Agricultural Chemistry and Biochemistry, The University of Agriculture, Peshawar, Pakistan, 7 Department of Clinical Laboratory Sciences, The Faculty of Applied Medical Sciences, Taif University, Taif, Saudi Arabia, 8 Research Center for Health Sciences, Deanship of Graduate Studies and Scientific Research, Taif University, Taif, Saudi Arabia, 9 Chair and Department of Medical Chemistry and Laboratory Medicine, Poznan University of Medical Sciences, Poznan, Poland,

* mashaljcs@yahoo.com (AR); rahaf.ajaj@adu.ac.ae (RA)

## Abstract

Elevated β-glucuronidase activity is associated with the production of toxic metabolites that contribute to tumor development and other diseases. Inhibiting this enzyme may offer therapeutic potential, including the prevention of colonic carcinogenesis. This study investigates the antidiabetic potential of metabolites derived from *Fernandoa adenophylla*, using β-glucuronidase as a model enzyme linked to hyperglycemia. Both *Escherichia coli* and human isoforms of β-glucuronidase were evaluated. Among the tested compounds, AA and DD exhibited the most significant inhibitory activity against the *E. coli* isoenzyme, with inhibition rates of 85.2% (IC$_{50}$ = 12.3 µM) and 82.6% (IC$_{50}$ = 8.2 µM), respectively. Against the human isoenzyme, compounds DD and CC showed the highest inhibition, with 92.6% (IC$_{50}$ = 28.2 µM) and 90.4% (IC$_{50}$ = 8.9 µM), respectively. These findings were further supported by molecular docking and molecular dynamics simulations. So, these results highlight the potential of *F. adenophylla* metabolites as promising candidates for developing novel therapeutic agents targeting β-glucuronidase.

**Data availability statement:** All relevant data are within the manuscript and its Supporting Information files.

**Funding:** This research was funded by Taif University, Saudi Arabia, Project No. (TU-DSPP-2024-09).

**Competing interests:** All the authors declare no conflicts of interest.

## 1. Introduction

β-glucuronidase is a lysosomal enzyme found in mammals, bacteria, and other organisms, where it plays a key role in the hydrolysis of β-d-glucuronides. In humans, this enzyme is present in various tissues, including the colon and human milk, and contributes to the reabsorption of substances like bilirubin and estrogens. Altered β-glucuronidase activity has been associated with several pathological conditions, including cancers (e.g., breast, colon), diabetes mellitus (DM), rheumatoid arthritis (RA), and infections [1–6]. Its role in tumor development, inflammation, and metabolic disorders makes β-glucuronidase an attractive target for therapeutic interventions, particularly in cancer treatment, neonatal jaundice, diabetes management, and anti-inflammatory therapies.

Due to the broad properties of this enzyme, inhibition of its activity has become a target for diagnostic and therapeutic applications in (a) anticancer chemotherapy - due to the enzyme's role in tumor development and metastasis; (b) neonatal jaundice treatment - the enzyme is highly expressed in human milk and has been implicated in the hyperbilirubinemia development; (c) diabetes mellitus management, due to the demonstrated relationship between disease status and enzyme activity levels, as well as concomitant periodontitis [7]; and (d) anti-inflammatory treatment, due to its pro-inflammatory role following extensive release from degranulated mast cells and neutrophils. Inhibition of β-glucuronidase alleviated these states and their side effects, thereby improving the efficacy of treatment regimens [1].

Some natural products and synthetic scaffolds have shown potential as β-glucuronidase inhibitors, with a few making their way into clinical use despite their moderate to weak pharmacokinetic profiles. Plants, long revered for their role in producing bioactive compounds, have been extensively studied for their medicinal properties throughout human history. Recently, these natural compounds are being explored as novel pharmaceutical scaffolds for developing modern drugs, a trend that has been well-documented [8]. With the increasing interest in plant-based substances and herbal remedies, they are emerging as a preferred alternative to synthetic pharmaceuticals, offering cost-effectiveness and a reduced likelihood of adverse effects. This shift is particularly significant given the high costs and undesirable side effects often associated with synthetic drugs and the growing concern about their role in developing antibiotic resistance in bacterial pathogens [9].

Identifying novel therapeutic applications from medicinal plants remains a critical aspect of pharmaceutical research. Ethnobotanical studies have played an essential role in discovering and utilizing these plant species [10]. The literature extensively covers the diverse range of plant-based medicines and isolated compounds investigated for their therapeutic potential [11]. It is also important to highlight that the indigenous knowledge held by local communities regarding the medicinal properties of plants has been instrumental in developing many pharmaceutical drugs [12]. The pharmaceutical industry has greatly benefited from phytochemical, biological, and pharmacological studies [13].

Numerous plant species with potential health benefits have already been scientifically investigated. However, many plants remain under-researched, leaving gaps in understanding their potential therapeutic applications. The primary objective of this study was to investigate and evaluate the medicinal properties of *Fernandoa adenophylla*, a species of the Bignoniaceae family that occurs in regions of Africa and Southeast Asia [14]. Various phytochemicals in this plant have been well documented in scientific literature. Key compounds include tecomoquinone-I, β-amyrin, lapachol, α-lapachone, β-sitosterol, adenophylone, dehydro-α-lapachone, and dehydro-iso-α-lapachone [15]. These compounds are of particular interest due to their potential as β-glucuronidase inhibitors, which is the focus of our research.

*Fernandoa adenophylla* has been traditionally used by local communities for various medical purposes, including managing amenorrhea, premature ejaculation, diabetes, night emissions, antiseptic treatments, antimicrobial effects, and skin diseases [16]. It is noteworthy that all species within the Bignoniaceae family are known to produce lapachol in significant quantities. The extensive literature on lapachol and its derivatives highlights their broad spectrum of biological activities. Compounds such as lapachol, lapachone, indanone derivatives, and peshawarquinone have been reported for their notable anti-inflammatory, analgesic, muscle relaxant, phosphodiesterase-1 inhibitory, and sedative properties [17–19].

This study focuses on evaluating the β-glucuronidase inhibitory activity of five compounds isolated from *Fernandoa adenophylla*. The rationale for selecting this plant stems from its documented phytochemical profile and its traditional use in managing conditions linked to β-glucuronidase activity. Additionally, molecular docking studies were employed to further investigate the mechanism of inhibition, providing insights into the therapeutic potential of these compounds..

## 2. Materials and methods

### 2.1. Ethics

Not applicable

### 2.2. General Procedure

The glass chromatography column used was 6 cm in diameter and 55 cm in length. Approximately 90 g of silica (200–300 mesh, Merck, Germany) was loaded into the column as the stationary phase, and varying combination ratios of mobile phase, chloroform, and methanol were utilized to isolate these compounds. Mass spectra were carried out on a JEOL MS Route Direct Probe. The $^1$H and $^{13}$C NMR and HMBC spectra were recorded on a Bruker Avance-AV- 400 and 100 MHz at the Hussain Ejaz Research Center of Chemistry, University of Karachi, Pakistan. Two-dimensional (2D) (HSQC, COSY, and HMBC) studies were performed using Topspin software. All commercial-grade solvents were distilled before use for extraction and column chromatography.

### 2.3. Plant collection

The heartwood roots of *Fernandoa adenophylla* were collected in April 2011 from the University of Peshawar, Khyber Pakhtunkhwa, Pakistan. The plant was identified by Professor Dr. Barkath Ullah, a distinguished researcher from the Department of Botany at the University of Peshawar. A voucher specimen, labeled with the unique identifier UOP/Bot987, has been deposited in the herbarium for future reference and research purposes.

### 2.4. Isolation and extraction

The heartwood roots of *Fernandoa adenophylla* were subjected to a shade-drying process for approximately 40–45 days. The dried material was pulverized into a fine powder and subsequently underwent extraction in a soxhlet apparatus using methanol as the solvent for 6 hours. The crude methanolic extract was subjected to sequential fractionation based on the increasing polarity of the solvents used. Following the process of fractionation and subsequent concentration on a rotary evaporator under reduced pressure, the fractions obtained from n-hexane, dichloromethane, ethyl acetate, and methanol

resulted in extract yields of 8 g, 15 g, 25 g, and 40 g, respectively. Column chromatography (CC) was conducted on the methanol fraction (25 g) utilizing silica gel (70–230 mesh) as stationary phase and a chloroform: methanol (10:0→0:10) mobile phase. The procedure yielded 114 fractions, which were subsequently consolidated into three main fractions based on TLC profiles: 1 (10.0→8:2, 600 mL, 6.2 g), 2 (8:2→7:3, 1500 mL, 12 g), and 3 (7:3→3:7, 1500 mL, 3.5 g). Fraction 1 on loading to a column having silica as stationary phase and chloroform (10:0, 1000 mL) yielded Compound (**5**) (30.42 mg). Fraction 2 (3.5 g) underwent further CC with chloroform and methanol (10:0→6:4), resulting in two subfractions: 2.1 (10.0→7:3, 600 mL, 1.5 g) and 2.2 (7:3→6:4, 1500 mL, 3 g). The fraction 2.1 was loaded to the small column on silica and eluting solvent chloroform and methanol (10:0→8:2, 400 ml), yielding compound (**4**) (14.32 mg). CC of fraction 2.2 using chloroform: methanol (6:4) produced compound (**1**) (12.56 mg). Fraction 3 (3.5 g) was subjected to CC on silica with chloroform: methanol (10.0→7:3), generating three sub-fractions: 3.1 (10.0→7:3, 300 mL, 1.5 g), 3.2 (7:3→5:5, 700 mL, 2.3 g), and 3.3 (5:5→2:8, 400 mL, 0.5 g). The sub-fraction 3.1 on separation through the column using a solvent in the ratio (8:2→5:5, 800 mL) yielded compound (**3**) (9.27 mg). Subsequently, CC of sub-fraction 3.2 (2.3 g) on silica gel using chloroform: methanol (7:3) yielded compound (**2**) (8.10 mg). The compounds (1–5; **Fig 1**) underwent additional purification by washing with n-hexane. The chemical structures of isolated compounds was characterized by spectroscopic data (See S1 File) [14].

## 2.5. β-glucuronidase inhibitory assay

The present study investigated the inhibitory potential of five isolated compounds from *Fernandoa adenophylla* against β-glucuronidase activity. The evaluation of these compounds for their inhibitory effects on β-glucuronidase was conducted following a well-documented protocol available in the literature [20]. The assessment of the inhibitory activity of compounds (**1–5**) against β-glucuronidase was carried out by monitoring the absorbance of p-nitrophenol, which is generated from the hydrolysis of p-nitrophenyl-b-D-glucuronide (N-1627), at a specific wavelength of 405 nm utilizing a spectropho-tometer. The experimental procedure involved the utilization of a reaction mixture consisting of 185 µL of a 0.1 M acetate buffer, 5 µL of a test compound solution (dissolved in 100% DMSO), and 10 µL of an enzyme solution (prepared using the buffer above at a concentration of 1 U/well or 1 U/250 µL). This reaction mixture was subjected to incubation at a temperature of 37 °C for thirty minutes. Subsequently, a volume of 50 µL of a solution containing p-nitrophenyl-β-D-glucuronide at a concentration of 0.4 mM was meticulously introduced into each well. The plates were then analyzed using a multiple reader, specifically the SpectraMax plus 384 model manufactured by Molecular Devices, located in San Jose, CA, USA. The readings were obtained at a wavelength of 405 nm. The experimental protocol was replicated thrice for each compound. The inhibitor used in this study to inhibit ß-glucuronidase was D-saccharic acid 1,4-lactone, as reported by Iqbal et al. in 2022 [21]. The IC$_{50}$ values were determined by using the following equation 1: (See S1 File).

$$\%Inhibition = 100 - \left( \frac{O.D_{test\ compound}}{O.D_{control}} \right) \times 100$$

(1)

## 2.6. Molecular docking

The generation of three-dimensional structures for isolated compounds from *Fernandoa adenophylla* was accomplished using the MOE software and all structures were energy-minimized [22]. The compounds underwent protonation, minimization, and charge assignment procedures following the initial synthesis steps. The three-dimensional crystal structures of the target proteins, β-glucuronidase from *Escherichia coli* and human β-glucuronidase, were obtained from the Protein Data Bank (PDB IDs: 6LEL and 1BHG) [23,24]. The structure analysis revealed that β-glucuronidase forms a homodimer consisting of chains A and B. Chain A was extracted for further analysis, and missing atoms and bond and angle corrections were addressed using the auto-correction function in MOE.

Fig 1. Chemical structures of compounds isolated from *Fernandoa adenophylla.*

To prepare the protein for docking simulations, it was subjected to protonation, charging, and minimization protocols to optimize its structure. Active site information was gathered from existing literature, allowing for precise targeting of critical residues responsible for enzyme inhibition [25]. Docking simulations of the newly isolated compounds were performed using default parameters, with two distinct rescoring functions applied: GBVI/WSA dG and the Triangle Matcher method. Thirty conformations were generated for each compound to identify the optimal binding conformation.

The results from the docking experiments were systematically organized and saved in output files in the MDB (Microsoft Access Database) format. These output files were visually examined to analyze the protein-ligand interactions, focusing on the active site of the target proteins. This analysis aimed to elucidate the inhibitory activity of the compounds isolated from *Fernandoa adenophylla*.

## 2.7. Molecular dynamics (MD) simulation

Combining docking results with MD simulations validates the docking predictions by assessing protein-ligand complexes' conformational flexibility and structural stability. In this study, MD simulations were performed using Amber [26] to analyze the dynamic behavior of the protein-ligand complexes over time. These simulations provide insight into molecular motions and interactions within a simulated environment.

The force field used in MD simulations is crucial for computing the potential energy of the protein-ligand complexes [27]. In this work, the General AMBER Force Field (GAFF) was applied to the ligands, while the FF14SB AMBER force field was used for the protein. Topology and coordinate files for each system were generated using the leap module of AMBER software. Atomic charges and topology files for the ligands were created using the antechamber suite in AMBER.

Solvation is an essential step for studying the internal motion of the protein at various temperatures. A truncated octahedral box of TIP3P water molecules was used to solvate the system, with water molecules added using the LEap module of Amber. Sodium ions (Na+) were introduced to neutralize the system. The Particle Mesh Ewald (PME) method was employed to compute long-range electrostatic interactions, and a cut-off distance of 10 Å was set for non-bonded interactions. The SHAKE algorithm was applied to constrain hydrogen-containing bonds.

The simulation protocol involved heating the system from 0 to 300 K, followed by equilibration at constant pressure and 300 K temperature. A production run of 100 ns was then performed for the compound 1 complex and the reference compound [28].

## 2.8. Statical analysis

ANOVA software was used for statistical analysis. The results of three experiments were presented as mean values ± SEM.

## 3. Results

### 3.1. β-glucuronidase inhibitory assay

The results from the in vitro evaluation of β-glucuronidase isozyme inhibitory activity (using E. coli and human sources) for the isolated derivatives (1–5) of *Fernandoa adenophylla* are presented in Table 1. The highest inhibitory effects for the E. coli isozyme were observed with compounds AA and DD, showing inhibition rates of 85.2% ($IC_{50}$ = 12.3 μM) and 82.6%

**Table 1. In vitro activity of the isolated compounds against β-glucuronidase isozymes.**

| Compounds | E. coli β-glucuronidase | | Human β-glucuronidase | |
|---|---|---|---|---|
| | % Inhibition | $IC_{50}$ ± SEM | % Inhibition | $IC_{50}$ ± SEM |
| AA (Lapachol; 1) | 85.2 | 12.3 ± 0.4 | 14.6 | NA |
| DD (Alpha-lapachone; 2) | 82.6 | 8.2 ± 0.5 | 87.1 | 28.2 ± 0.3 |
| EE (Peshawaraquinone; 3) | 41.1 | NA | 92.6 | 32.2 ± 0.7 |
| BB (Dehydro-α-lapachone; 4) | 72.6 | 37.6 ± 0.5 | 20.9 | NA |
| CC (Indanone derivatives; 5) | 71.6 | 40.2 ± 0.8 | 90.4 | 8.9 ± 0.9 |
| D-Saccharic acid 1,4-lactone | 88.9 | 42.7 ± 0.7 | 88.9 | 38.1 ± 0.6 |

(IC$_{50}$=8.2 µM), respectively. In contrast, the human isozyme was significantly inhibited by compounds EE (92.6%) and CC (90.4%), with IC$_{50}$ values of 32.2 µM and 8.9 µM, respectively. The performance of the positive control drug was remarkable, demonstrating superior efficacy compared to the isolated compounds.

## 3.2. Molecular docking analysis

Molecular docking was performed using MOE software to explore the binding interactions of compounds isolated from *Fernandoa adenophylla* with the target enzyme β-glucuronidase. The isolated compounds were docked into the active sites of both E. coli and human β-glucuronidase. The docking results revealed that these compounds could be effectively accommodated within the β-glucuronidase binding pocket, see Table 1.

For the E. coli β-glucuronidase, the reference compound (D-Saccharic acid 1,4-lactone) displayed one conventional hydrogen bonds with TYR472, and two carbon hydrogen interactions with SER360, achieving a docking score of -6.699 kcal/mol.

Indanone derivatives demonstrated one conventional hydrogen bond with ASN 358, one carbon-hydrogen bond with LEU361, one pi-pi stacked interaction with TYR472, and six alkyl and pi-alkyl interactions with LEU361, TYR468, TYR472, LEU561 residues, resulting in a docking score of -7.2419 kcal/mol. Lapachol, showed two conventional hydrogen bond interactions with LEU361 and MET447, six alkyl and pi-alkyl interation with LEU361, VAL446, MET447, TYR468, TYR472, TRP549, LYS568 residues, yielding a docking score of -7.210. The third potent compound, alpha-apache, with a docking score of -7.197 kcal/mol, formed one conventional hydrogen bond with MET447, and seven TYR468, TYR472, TRP549, and LYS568, residues. For details, see (Table 2, Fig 2 and Fig 3).

In the case of human β-glucuronidase, the Indanone derivatives emerged as the top compound based on interactions and docking score (-6.887 kcal/mol), forming four conventional hydrogen bonds with HIS94, TYR199 and GLN202, one pi-sigma interaction with GLN202, three carbon hydrogen interactions with PHE200, VAL201 and three alkyl interactions with HIS94, VAL96, TYR199. The second most potent compound was alpha-lapachone, followed by Lapachol, with docking scores of -6.493 and -6.347, respectively. alpha-lapachone formed two conventional hydrogen bonds with TYR199 and GLN202, two pi-pi stacked interactions with HIS94, and one alkyl interaction with VAL201, while Lapachol established one conventional hydrogen bond acceptor interaction with PHE200, one pi-pi stacked interaction with TYR199 and three alkyl interactions with VAL96, VAL201 residues. For details, see Table 3 Fig 4, and Fig 5.

## 3.3. Molecular dynamic (MD) simulation and its analysis

The molecular dynamics (MD) simulation was conducted to assess the stability of the most potent compound compared to the reference compound. The trajectory analysis was performed using the CPPTRAJ module of AMBER software. Subsequent post-simulation analyses, including RMSD and RMSF, were carried out to evaluate and compare the stability and flexibility of the protein-ligand complexes.

**Table 2. Illustrated the binding score and interacting residues of E.coli β-glucuronidase involved in binding interaction.**

| No | Compounds | Residues | S-Score |
|---|---|---|---|
| 1 | AA (Lapachol; 1) | LEU361, VAL446, MET447, TYR468, TYR472, TRP549, LYS568 | -7.210 |
| 2 | DD (Alpha-lapachone; 2) | MET447, TYR468, TYR472, TRP549, LYS568 | -7.197 |
| 3 | EE (Peshawaraquinone; 3) | PHE448, TYR472 | -6.483 |
| 4 | BB (Dehydro-α-lapachone; 4) | LEU361, PHE448, TYR468, TYR472 | -6.594 |
| 5 | CC (Indanone derivatives; 5) | ASN358, LEU361, TYR468, TYR472, LEU561 | -7.241 |
| 6 | D-Saccharic acid 1,4-lactone | SER360, TYR472 | -6.699 |

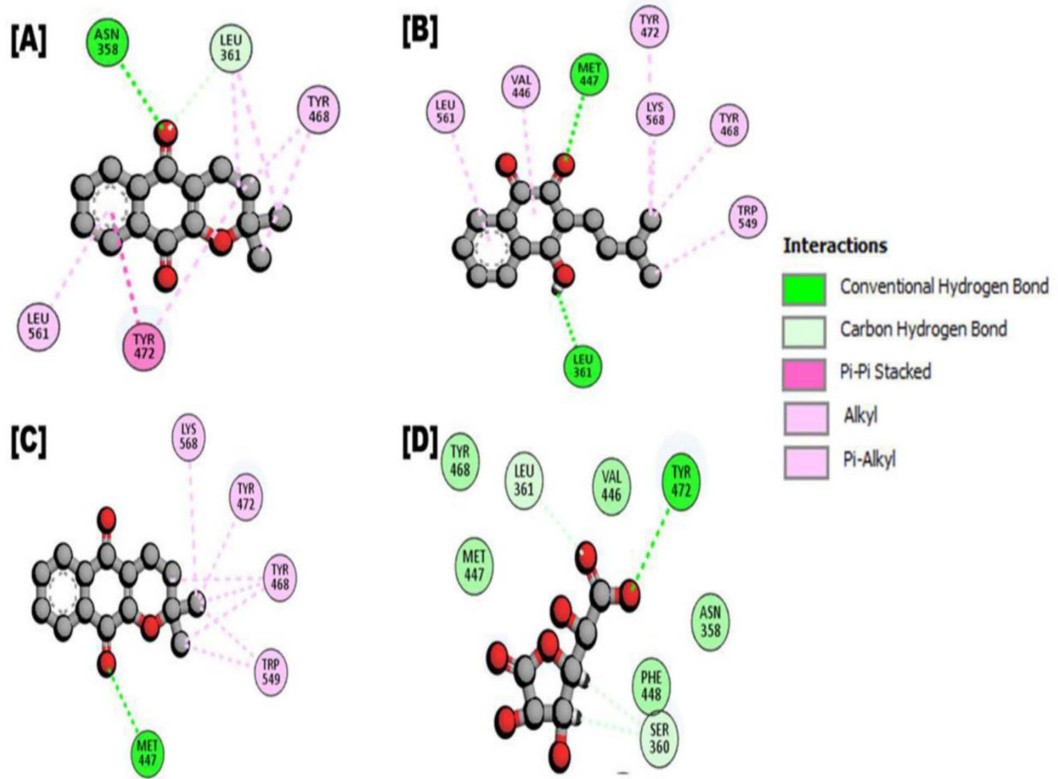

**Fig 2. Represented the 2d interaction of protein-ligand (*Fernandoa adenophylla* isolated compounds in complex with E coli β-glucuronidase).** A Interaction of Indanone derivatives; **B.** Ligand interaction of Lapachol; **C.** Ligand interaction of Alpha-lapachone; **D.** Ligand interaction of D-Saccharic acid 1, 4-lactone (Reference compound).

### 3.4. Root mean square deviation

Root mean square deviation (RMSD) measures the differences in the backbone structure of protein complexes from their initial to final conformations. A larger RMSD curve indicates lower stability, while a smaller curve suggests higher stability. The MD simulation results demonstrated that the reference compound, D-saccharic acid 1,4-lactone, exhibited a more significant deviation than the most potent compound, the Indanone derivatives. Initially, the compound 1/eβ-glucuronidase complex RMSD was found to be∼0.4Å, which gradually increased and reached ~1.4Å with minor increases and decreases and remained stable until the end of the simulation, indicating a dynamically stable interaction (Fig 6). In contrast, the RMSD of the D-Saccharic acid 1,4-lactone/eβ-glucuronidase complex showed a stable pattern with a score of ~0.8Å from the beginning to 19ns, and then gradually increased to 1.4Å by 60ns. After 60ns, the RMSD, again gradually increased and then reached ~1.9Å. Then remained stable after 65ns until the end of the simulation. These results indicate that the compound 1/eβ-glucuronidase complex is more stable than the D-Saccharic acid 1,4-lactone/eβ-glucuronidase complex, as evidenced by its lower RMSD values and more stable behavior.

### 3.5. Root mean square fluctuation

Root mean square fluctuations (RMSF) provide insights into the flexibility of residues within a protein-ligand complex. Regions with low RMSF values indicate rigidity, while higher RMSF values suggest increased flexibility. The RMSF analysis in this study revealed that, compared to the reference compound (D-Saccharic acid 1,4-lactone),

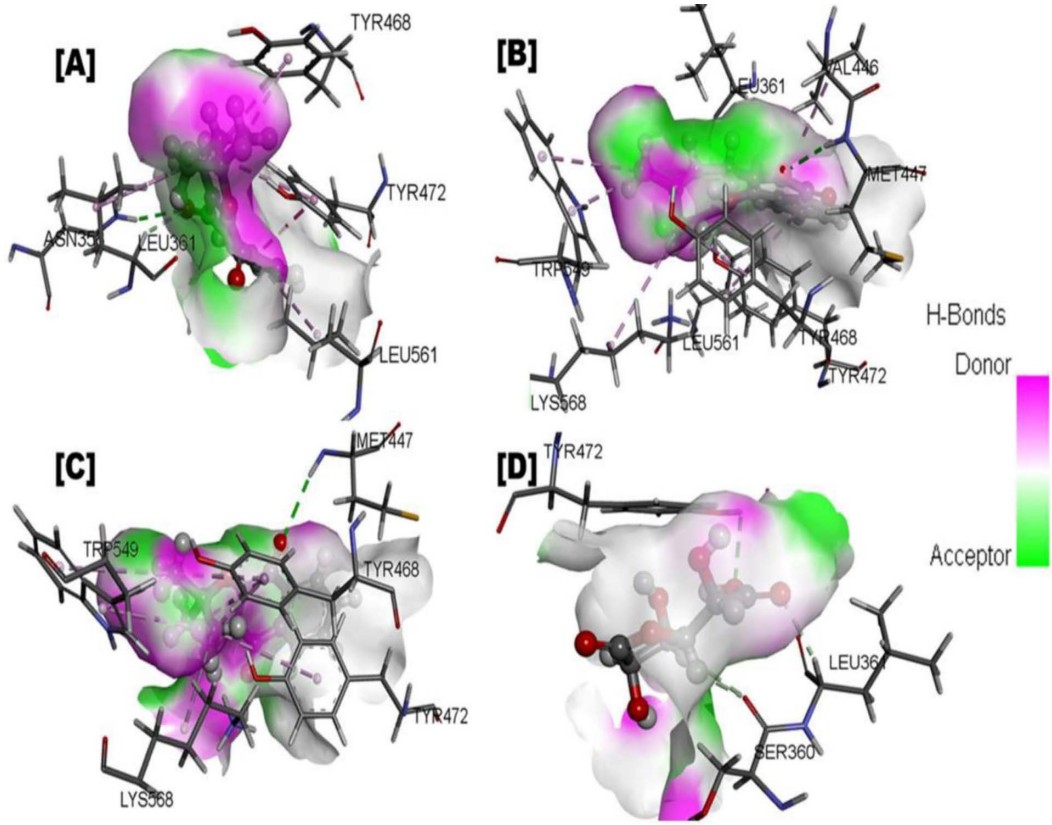

**Fig 3. Binding interaction of protein-ligand (*Fernandoa adenophylla* isolated compounds in complex with E coli β-glucuronidase).** A Interaction of Indanone derivatives; **B.** Ligand interaction of Lapachol; **C.** Ligand interaction of Alpha-lapachone; **D.** Ligand interaction of D-Saccharic acid 1, 4-lactone (Reference compound).

**Table 3. Illustrated the binding score and interacting residues of human β-glucuronidase involved in binding interaction.**

| No | Compounds | Residues | S-Score |
|---|---|---|---|
| 1 | AA (Lapachol; 1) | VAL96, TYR199, PHE200, VAL201 | -6.347 |
| 2 | DD (Alpha-lapachone; 2) | HIS94, TYR199, VAL201, GLN202 | -6.493 |
| 3 | EE (Peshawaraquinone; 3) | SER52, VAL96, TYR199 | -6. 094 |
| 4 | BB (Dehydro-α-lapachone; 4) | ARG91, PHE95, VAL96, TYR199 | -5.977 |
| 5 | CC (Indanone derivatives; 5) | HIS94, VAL96, TYR199, PHE200, VAL201, GLN202 | -6.887 |
| 6 | D-Saccharic acid 1,4-lactone | TYR199, GLN202 | -6. 098 |

the compound 1/eβ-glucuronidase complex exhibited lower RMSF values, indicating enhanced stability. Specifically, residues 140–160, 220–260, 360–370, 550–555 of the reference compound showed more significant fluctuations compared to the compound 1/eβ-glucuronidase complex. Both systems, however, maintained an average RMSF of ~0.5 Å, with notable fluctuations observed in the 140–160, 220–260, 360–370 and 550–555 regions. Although the reference compound exhibited higher fluctuations, the overall residual flexibility pattern was similar to that of the 1/eβ-glucuronidase complex.

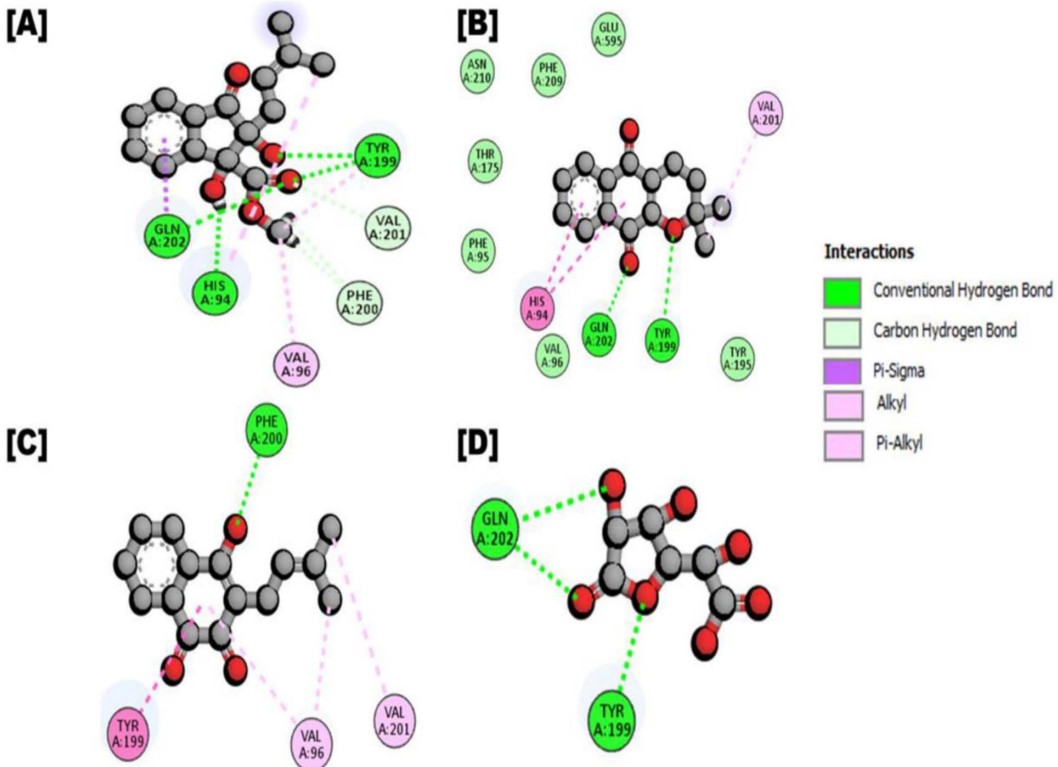

**Fig 4. Represented the 2d interaction of protein-ligand (*Fernandoa adenophylla* isolated compounds complex with human β-glucuronidase).** A Interaction of Indanone derivatives; **B.** Ligand interaction of alpha-lapachone; **C.** Ligand interaction of Lapachol; **D.** Ligand interaction of D-Saccharic acid 1, 4-lactone (Reference compound).

As illustrated in Fig 7, the RMSF profiles clearly show the differences in flexibility between the two systems. These findings suggest that the binding of the selected inhibitor to the target protein enhances its stability, as evidenced by the reduced fluctuations in critical regions.

### 3.6. Distance

The AMBER software was used to calculate the distance between the eβ-glucuronidase and ligands during the simulation. The average distance calculations were done using the 100 ns MD trajectories and are shown in Fig 8. The average distance for the compound 1/eβ-glucuronidase complex was observed to be around 4.25 nm, while for D-Saccharic acid 1,4-lactone/eβ-glucuronidase complex, it was computed to be around 4.38 nm. Distance analysis shows that compound-1 has the same distance from eβ-glucuronidase as the reference D-Saccharic acid 1,4-lactone.

### 3.7. Binding Free energy (BFE) calculation.

The binding free energy of each complex is predicted using a widely used method called MM-GBSA. All energy terms are listed in Table 4, including vdWaals (vdW), electrostatic energy (EEL), polar solvation (EGB), and total binding free energy of all systems. The total binding energy of the compound 1/eβ-glucuronidase complex was found to be –26.18 kcal/mol. The vdW value reported for the compound 1/eβ-glucuronidase complex was -26.18 kcal/mol, electrostatic -85.88 kcal/mol, EGB 73.21, and ESURF -3.10 kcal/mol. Total binding energy for reference/eβ-glucuronidase complex was found to be

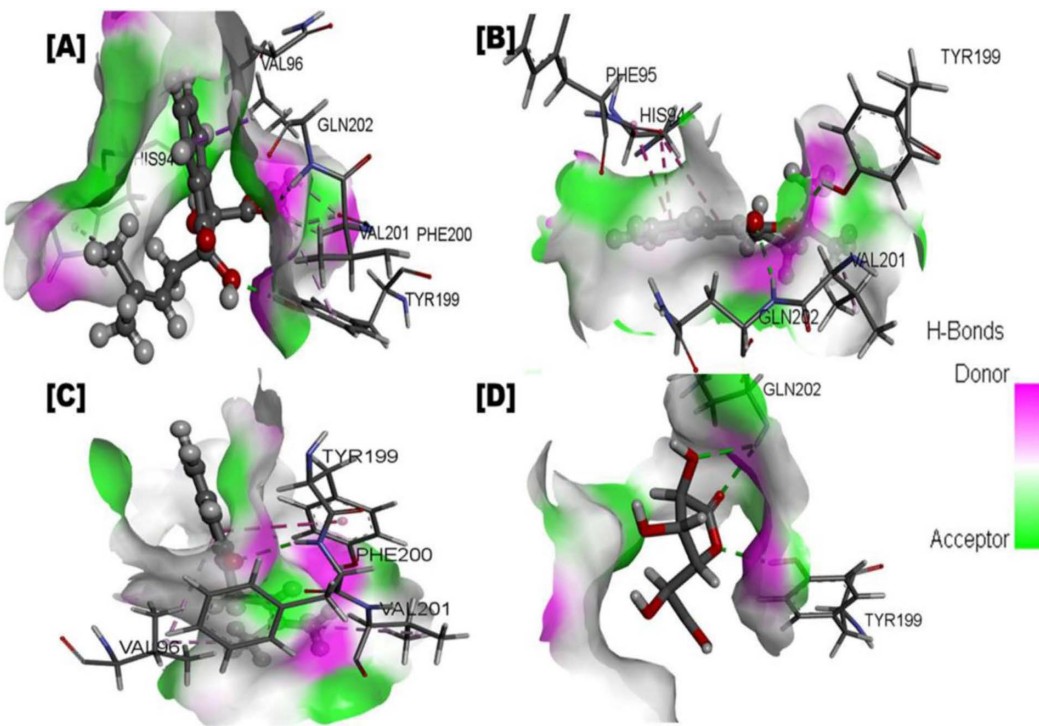

**Fig 5. Binding interaction of protein-ligand (*Fernandoa adenophylla* isolated compounds complex with human β-glucuronidase). A** Interaction of Indanone derivatives; **B.** Ligand interaction of alpha-lapachone; **C.** Ligand interaction of Lapachol; **D.** Ligand interaction of D-Saccharic acid 1, 4-lactone (Reference compound).

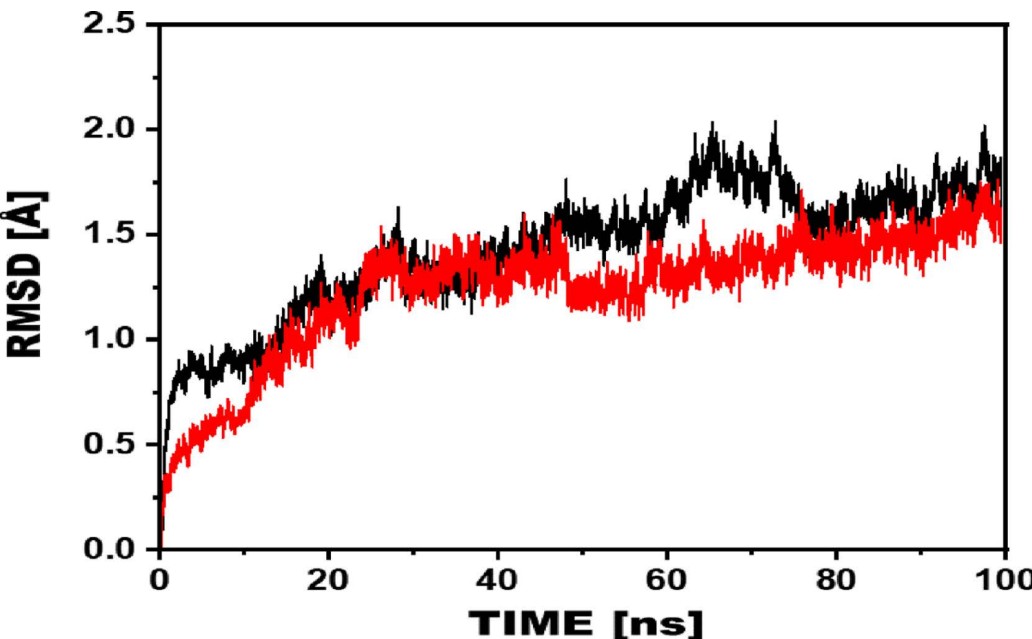

**Fig 6. RMSD of the compound 1/eβ-glucuronidase complex is shown in red, while the reference compound complex is shown in black.**

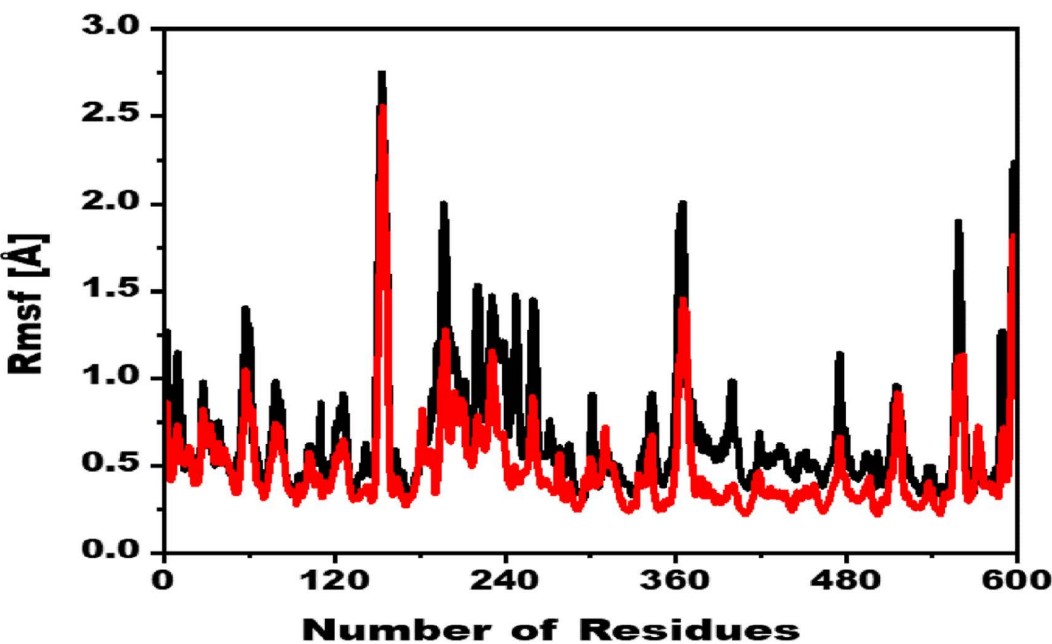

**Fig 7. Root means square fluctuations (RMSF) of D-Saccharic acid 1, 4-lactone, and Lapachol during 100ns MD simulations.**

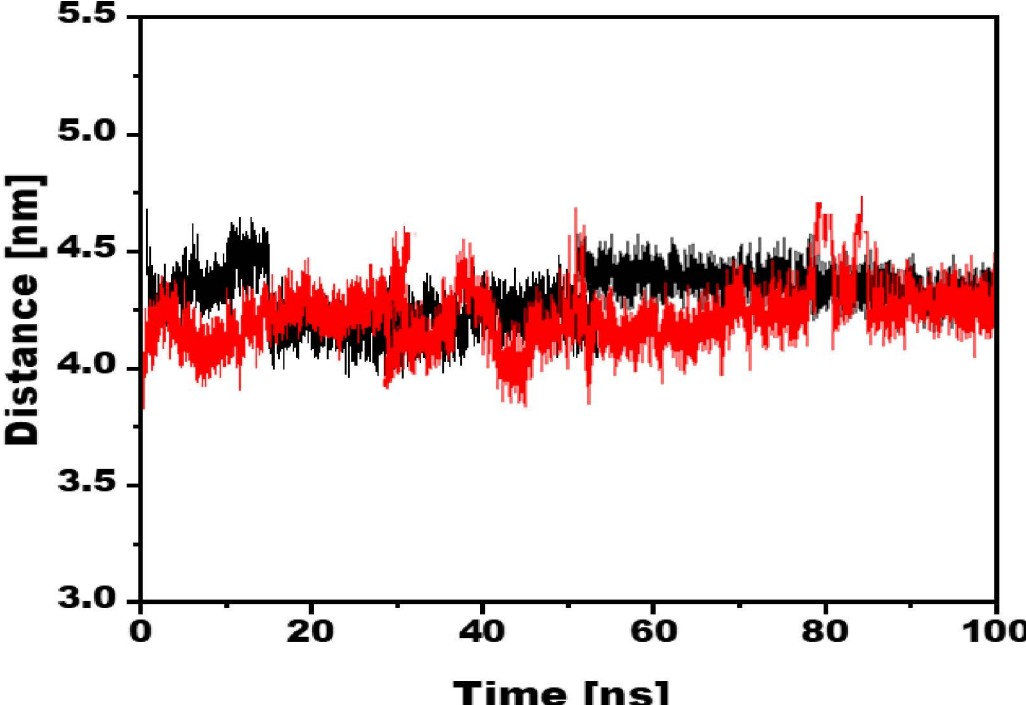

**Fig 8. The distance of the compound 1/e β-glucuronidase complex is shown in red, while the reference compound complex is shown in black, during 100ns MD simulations.**

**Table 4. The total binding free energy and related term of compound-1/eβ-glucuronidase complex and reference/eβ-glucuronidase complex.**

| S.NO | System | VDWAALS | EEL | EGB | ESURF | DELTA TOTAL kcal/mol |
|------|--------|---------|-----|-----|-------|----------------------|
| 1. | Compound-1 | -10.4084 | -85.8818 | 73.2107 | -3.1078 | -26.1873 |
| 2. | Reference | -24.0952 | -11.4873 | 20.2630 | -2.6843 | -18.0002 |

-18.00. For reference/eβ-glucuronidase complex, the calculated values of vdW and electrostatic energy, EGB and ESURF were -24.09, -11.48 kcal/mol, 20.26, and -18. 00 kcal/mol, respectively. Table 4 shows the total binding energy along with all related terms.

## 4. Discussion

Natural products continue to be valuable sources for drug discovery, especially for chronic diseases like diabetes mellitus (DM). These compounds have demonstrated various bioactivities, including anti-inflammatory, antioxidant, and antimicrobial properties, making them valuable candidates for treating chronic conditions. Among these, DM has garnered significant attention due to its increasing prevalence and the limitations of current synthetic therapies. In searching for safer and more effective treatments, natural products offer promising alternatives for managing DM and related metabolic disorders. DM and obesity stand as one of the most prevalent chronic health challenges worldwide [29–30]. The management of DM primarily relies on synthetic drugs, especially in developing countries, while monthly depot injections like semaglutide are available in more developed regions [31]. However, there remains an urgent need for safer, more effective, and cost-efficient antidiabetic treatments. The exploration of natural products [32] and their isolated compounds [33] continues to be a promising area of research in this quest. This study was undertaken to discover a potent and novel antidiabetic remedy.

Previous research has reported the antidiabetic potential of *Fernandoa adenophylla* [32]. Given the promising results of the plant extract, we conducted a detailed investigation into its chemical constituents. β-glucuronidase, an enzyme essential for human metabolism, is critical in carbohydrate breakdown [34]. However, the excessive activity of this enzyme is linked to the release of free glucose, which can lead to hyperglycemia and, ultimately, type 2 diabetes mellitus. The pathological accumulation of glucosylceramide, resulting from β-glucuronidase deficiency, is also associated with Gaucher's disease [35,36]. Therefore, inhibiting β-glucuronidase has become a key strategy in managing type 2 diabetes mellitus (DM2).

In our study, the isolated compounds from *Fernandoa adenophylla* showed substantial inhibitory activity against β-glucosidase, supporting their potential antidiabetic properties. The compounds exhibited powerful binding behavior to the target enzyme through molecular docking tests together with substantial interface connections which supported their inhibitory mechanisms. Molecular dynamics simulations proved the stability of these interactions during a 100-nanosecond simulation run. Without extending enzyme kinetics analysis more certain confirmation about the specific binding site on β-glucuronidase cannot be achieved since multiple ligand-binding sites exist. Enzyme kinetics experiments represent the essential research step for future validation because they will confirm the specific binding site and substantiate the docking study findings. The top-performing compound showed significant structural stability and consistent binding, suggesting its potential as a promising lead molecule for further development. Our findings indicate that the antidiabetic effects of *Fernandoa adenophylla* may stem from its β-glucosidase inhibitory activity, which could effectively reduce glucose levels and mitigate hyperglycemia. The combined in vitro and in silico results support the therapeutic potential of these compounds and provide a foundation for future research aimed at developing natural, enzyme-targeting antidiabetic drugs.

## 5. Conclusions

In the present study, the isolated chemical constituents (**1–5**) of *Fernandoa adenophylla* exhibited remarkable antidiabetic activity. This potency was reinforced by computational analyses, including molecular docking and preliminary molecular

dynamics simulations, which confirmed that these compounds formed stable and effective interactions with the target enzymes. The combined results from both *in vitro* and *in silico* studies underscore the potential of these natural compounds as promising candidates for antidiabetic drug development.

## Supporting information

**S1 File. The spectroscopic data of all isolated compounds are available in the supplementary file.**
(DOC)

## Author contributions

**Conceptualization:** Abdur Rauf, Rahaf Ajaj.

**Formal analysis:** Abdur Rauf, Rahaf Ajaj.

**Funding acquisition:** Rahaf Ajaj, Abdulhakeem S. Alamri.

**Investigation:** Zuneera Akram.

**Methodology:** Majid Khan, Humaira Hussain, Dorota Formanowicz.

**Project administration:** Abdur Rauf, Abdul Wadood.

**Supervision:** Maryam Zulfat, Zafar Ali Shah.

**Validation:** Abdulhakeem S. Alamri, Walaa F. Alsanie, Majid Alhomrani.

**Writing – original draft:** Humaira Hussain.

**Writing – review & editing:** Dorota Formanowicz.

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
