## [Editor Report · Decision Letter 0]

12 Feb 2025

PONE-D-25-04376Investigation of the inhibitory potential of secondary metabolites isolated from Fernandoa adenophylla against Beta-glucuronidase via molecular docking and molecular dynamics simulation studiesPLOS ONE

Dear Dr. Rauf,

Thank you for submitting your manuscript to PLOS ONE. After careful consideration, we feel that it has merit but does not fully meet PLOS ONE’s publication criteria as it currently stands. Therefore, we invite you to submit a revised version of the manuscript that addresses the points raised during the review process.

**ACADEMIC EDITOR: ** In this study: " Investigation of the inhibitory potential of secondary metabolites isolated from Fernandoa adenophylla against Beta-glucuronidase via molecular docking and molecular dynamics simulation studies" The authors provide report about the inhibitory activities of compounds isolated from Fernandoa adenophylla against β-glucuronidase via in vitro and in silico studies. However, there are many concerns in this study, and major revision is required.

We look forward to receiving your revised manuscript.

Kind regards,

Viet Phong Nguyen, Ph.D.

Academic Editor

PLOS ONE

Journal Requirements:

Additional Editor Comments :

1. Abstract: Avoid non-standard or uncommon abbreviations. If any are essential to include, ensure they are defined within your abstract at first mention.

2. Section 2.1: - Details on chromatographic separation techniques are missing (type of chromatographic column, stationary/mobile phase used, elution conditions, detection).

- Details of spectroscopic techniques to determine the structure of the isolated compound (type of NMR spectroscopy, mass, equipment used).

3. Section 2.3: check again the writing here. There are many grammar and presentation issues.

Additionally, all NMR data along with Chemical properties and spectra are mandatory in Supplemental materials.

4. Section 2.5: references of enzyme structures downloaded from Protein Data Bank are required.

In addition, was a grid box defined around the binding site to specify the docking region, or was blind docking used in this study?

Please clearly indicate the grid box parameters (grid size, grid position) for each target protein.

How was the 3D structure of the inhibitor constructed? Was the structure of the test compounds minimized before the docking process? Please indicate.

5. Section 3.1: Please provide the IC50 graphs of all test compounds.

The β-glucuronidase enzyme has multiple ligand binding sites (e.g., catalytic and allosteric sites). Without conducting enzyme kinetics analysis, it is unclear which binding site is relevant for the inhibitors. Therefore, to validate the binding site and confirm the docking results, enzyme kinetics experiments are mandatory.

6. Section 3.2 Molecular docking analysis: There is much literature reporting the inhibition mode and binding site of β-glucuronidase inhibitors. Please compare your compounds with reported inhibitors. What is the difference between the binding sites? Please clearly describe.

7. Section 3.3 MD simulation: A molecular dynamics simulation of 20 ns is not meaningful and does not yield any significant scientific results. In fact, the protein-ligand complex tends to exhibit many fluctuations during the initial 20 ns equilibration of simulations. Therefore, a prolonged molecular dynamics simulation of at least 100 ns is required.

8. It would be beneficial to report the RMSD of the ligand after fitting to the protein backbone, along with the distance between the centers of mass of the protein and ligand. These parameters can reveal any potential internal motions, particularly interdomain motion, within the protein.

9. How about the binding affinity of bound ligand conformations? MM/PB(GB)SA calculations are required.
---

## [Author Response · Author response to Decision Letter 1]

17 Mar 2025

Dear Viet Phong Nguyen, Ph.D.

Academic Editor

PLOS ONE

Thank you very much for the reviewers’ comments concerning our manuscript. We have studied the reviewer comments carefully and have made several revisions to the text. We would like to express our appreciation to you and the reviewers for your many suggestions, which have greatly improved our manuscript. All changes are shown in yellow, highlighted in the revised manuscript, and are outlined below on a point-by-point basis (red color).

We hope these corrections and revisions are satisfactory and that the manuscript now meets the requirements for publication.

We look forward to hearing from you at your earliest convenience.

Journal Requirements:

Response: Done

Response: Done

Response: Done

Additional Editor Comments :

1. Abstract: Avoid non-standard or uncommon abbreviations. If any are essential to include, ensure they are defined within your abstract at first mention.

Response: Corrected as suggested.

2. Section 2.1: - Details on chromatographic separation techniques are missing (type of chromatographic column, stationary/mobile phase used, elution conditions, detection).

Response: Details on chromatographic separation techniques, including column specifications, stationary and mobile phases, elution conditions, and detection methods, have now been incorporated.

- Details of spectroscopic techniques to determine the structure of the isolated compound (type of NMR spectroscopy, mass, equipment used).

Response: Details of the spectroscopic data, along with the corresponding spectra, are provided in the supplementary file.

3. Section 2.3: check again the writing here. There are many grammar and presentation issues.

Additionally, all NMR data along with Chemical properties and spectra are mandatory in Supplemental materials.

Response: All grammatical issues have been addressed. The NMR data, physical data, and spectra are now provided in the Supplemental Materials.

Section 2.5: references of enzyme structures downloaded from Protein Data Bank are required. In addition, was a grid box defined around the binding site to specify the docking region, or was blind docking used in this study?

Response: Dear Reviewer, Thank you for your valuable comments. We have incorporated the suggested references and followed the docking procedure as reported in a previous study. (https://doi.org/10.3390/pr11030880, doi:10.5281/zenodo.3576583).

Please clearly indicate the grid box parameters (grid size, grid position) for each target protein. How was the 3D structure of the inhibitor constructed? Was the structure of the test compounds minimized before the docking process? Please indicate.

Response: The requested details have now been incorporated into the revised manuscript and have been highlighted for clarity.

Section 3.1: Please provide the IC50 graphs of all test compounds.

Response: We sincerely appreciate the reviewer's insightful comment. In response, we have incorporated the IC₅₀ graphs for all tested compounds (AA: Lapachol, DD: Alpha-lapachone, CC: Indanone derivatives, and D-Saccharic acid 1,4-lactone) as requested. The IC₅₀ values were determined using nonlinear regression analysis in GraphPad Prism, and the corresponding dose-response curves are now provided below.

The β-glucuronidase enzyme has multiple ligand binding sites (e.g., catalytic and allosteric sites). Without conducting enzyme kinetics analysis, it is unclear which binding site is relevant for the inhibitors. Therefore, to validate the binding site and confirm the docking results, enzyme kinetics experiments are mandatory.

Response: Thank you for your valuable comment. We acknowledge that β-glucuronidase has multiple ligand-binding sites, including catalytic and allosteric sites, and that enzyme kinetics experiments are essential for definitive validation. In our study, molecular docking was employed as an initial approach to predict potential interactions. However, to further substantiate our findings, we plan to conduct enzyme kinetics experiments in future studies to confirm the binding site and validate the docking results experimentally.

Section 3.2 Molecular docking analysis: There is much literature reporting the inhibition mode and binding site of β-glucuronidase inhibitors. Please compare your compounds with reported inhibitors. What is the difference between the binding sites? Please clearly describe.

Response: The compounds have now been compared with the reported inhibitors, and the updates are highlighted in the revised manuscript.

Section 3.3 MD simulation: A molecular dynamics simulation of 20 ns is not meaningful and does not yield any significant scientific results. In fact, the protein-ligand complex tends to exhibit many fluctuations during the initial 20 ns equilibration of simulations. Therefore, a prolonged molecular dynamics simulation of at least 100 ns is required.

Response: The MD simulation has been extended to 100 ns, and the updates have been highlighted in the revised manuscript.

It would be beneficial to report the RMSD of the ligand after fitting to the protein backbone, along with the distance between the centers of mass of the protein and ligand. These parameters can reveal any potential internal motions, particularly interdomain motion, within the protein.

Response: As per the suggestions, the RMSD and distance analysis have been incorporated and highlighted in the revised manuscript.

How about the binding affinity of bound ligand conformations? MM/PB(GB)SA calculations are required

Response: The required details have now been added and appropriately highlighted in the revised manuscript.

---

## [Decision Letter · Decision Letter 1]

9 Apr 2025

PONE-D-25-04376R1Investigation of the inhibitory potential of secondary metabolites isolated from Fernandoa adenophylla against Beta-glucuronidase via molecular docking and molecular dynamics simulation studiesPLOS ONE

Dear Dr. Rauf,

Thank you for submitting your manuscript to PLOS ONE. After careful consideration, we feel that it has merit but does not fully meet PLOS ONE’s publication criteria as it currently stands. Therefore, we invite you to submit a revised version of the manuscript that addresses the points raised during the review process.

We look forward to receiving your revised manuscript.

Kind regards,

Viet Phong Nguyen, Ph.D.

Academic Editor

PLOS ONE

Journal Requirements:

Reviewers' comments:

Reviewer's Responses to Questions

**Comments to the Author**

1. If the authors have adequately addressed your comments raised in a previous round of review and you feel that this manuscript is now acceptable for publication, you may indicate that here to bypass the “Comments to the Author” section, enter your conflict of interest statement in the “Confidential to Editor” section, and submit your "Accept" recommendation.

Reviewer #1: All comments have been addressed

Reviewer #2: All comments have been addressed

Reviewer #3: All comments have been addressed

2. Is the manuscript technically sound, and do the data support the conclusions?

Reviewer #1: Yes

Reviewer #2: Partly

Reviewer #3: Yes

3. Has the statistical analysis been performed appropriately and rigorously? 

Reviewer #1: N/A

Reviewer #2: Yes

Reviewer #3: (No Response)

4. Have the authors made all data underlying the findings in their manuscript fully available?

Reviewer #1: Yes

Reviewer #2: Yes

Reviewer #3: (No Response)

5. Is the manuscript presented in an intelligible fashion and written in standard English?

Reviewer #1: Yes

Reviewer #2: Yes

Reviewer #3: (No Response)

6. Review Comments to the Author

Reviewer #1: The authors adequately responded to all comments and suggestions raised to improve their manuscript.

Reviewer #2: In my opinion, the manuscript “Investigation of the inhibitory potential of secondary metabolites isolated from Fernandoa adenophylla against Beta-glucuronidase via molecular docking and molecular dynamics simulation studies” is well written and presents interesting information. But I would like to point out some issues which can be revised:

Abstract: Revise the abstract to align with conventional standards of clarity, logical flow, and objectivity.

Introduction: The introduction section should be rewritten in a more focused manner, in concise paragraphs. The statement of the problem and the rationale of this study is unclear. Although the authors have included some studies, it is needed to include additional literature on this plant, and a justification of its choice for the current investigation. According to this study, isolated chemical constituents (1-5) of Fernandoa adenophylla exhibited remarkable antidiabetic activity. Additionally, what are all the major phytocompounds have been reported and so on can be added.

Result: Appropriate inferential statistics are missing, especially for results. This is particularly important in interpreting the results, appraising their validity, and drawing appropriate conclusions.

Discussion : The discussion section should be revised and focused on interpretation, relationships with reports from other scholars in the field, and implication of the obtained results.

Reviewer #3: I recommend the acceptance of the manuscript. One additional comment I just mention that author should write the unit of binding energy in the manuscript.

7. PLOS authors have the option to publish the peer review history of their article (what does this mean? ). If published, this will include your full peer review and any attached files.

**Do you want your identity to be public for this peer review?** For information about this choice, including consent withdrawal, please see our Privacy Policy .

Reviewer #1: **Yes: ** Khaled M. Darwish

Reviewer #2: No

Reviewer #3: **Yes: ** Shashank Shekher Mishra

---

## [Author Response · Author response to Decision Letter 2]

14 Apr 2025

Dear Viet Phong Nguyen, Ph.D.

Academic Editor

PLOS ONE

Thank you very much for the reviewers’ comments concerning our manuscript. We have carefully reviewed the comments from the reviewers and made several revisions to the text. We would like to express our appreciation to you and the reviewers for your many suggestions, which have greatly improved our manuscript. All changes are highlighted in yellow in the revised manuscript and are outlined below on a point-by-point basis (in red).

We hope these corrections and revisions are satisfactory and that the manuscript now meets the requirements for publication.

We look forward to hearing from you at your earliest convenience.

Journal Requirements:

Response: The reference list has been checked and found to be corrected.

Reviewers' comments:

Reviewer's Responses to Questions

Comments to the Author

1. If the authors have adequately addressed your comments raised in a previous round of review and you feel that this manuscript is now acceptable for publication, you may indicate that here to bypass the “Comments to the Author” section, enter your conflict of interest statement in the “Confidential to Editor” section, and submit your "Accept" recommendation.

Reviewer #1: All comments have been addressed

Reviewer #2: All comments have been addressed

Reviewer #3: All comments have been addressed

Response: Thank you.

2. Is the manuscript technically sound, and do the data support the conclusions?

Reviewer #1: Yes

Reviewer #2: Partly

Reviewer #3: Yes

Response: The necessary corrections have been made.

3. Has the statistical analysis been performed appropriately and rigorously?

Reviewer #1: N/A

Reviewer #2: Yes

Reviewer #3: (No Response)

Response: Thank you.

4. Have the authors made all data underlying the findings in their manuscript fully available?

Reviewer #1: Yes

Reviewer #2: Yes

Reviewer #3: (No Response)

Response: Thank you.

5. Is the manuscript presented in an intelligible fashion and written in standard English?

Reviewer #1: Yes

Reviewer #2: Yes

Reviewer #3: (No Response)

Response: Thank you.

6. Review Comments to the Author

Reviewer #1: The authors adequately responded to all comments and suggestions raised to improve their manuscript.

Response: Thank you.

Reviewer #2: In my opinion, the manuscript “Investigation of the inhibitory potential of secondary metabolites isolated from Fernandoa adenophylla against Beta-glucuronidase via molecular docking and molecular dynamics simulation studies” is well written and presents interesting information. But I would like to point out some issues which can be revised:

Abstract: Revise the abstract to align with conventional standards of clarity, logical flow, and objectivity.

Response: Dear reviewer, thank you for your suggestion. We have revised the abstract to make it more concise and clear.

Introduction: The introduction section should be rewritten in a more focused manner, in concise paragraphs. The statement of the problem and the rationale of this study is unclear. Although the authors have included some studies, it is needed to include additional literature on this plant, and a justification of its choice for the current investigation. According to this study, isolated chemical constituents (1-5) of Fernandoa adenophylla exhibited remarkable antidiabetic activity. Additionally, what are all the major phytocompounds have been reported and so on can be added.

Response: Dear reviewer, thank you for your suggestion. The introduction has been revised as per the suggestions.

Result: Appropriate inferential statistics are missing, especially for results. This is particularly important in interpreting the results, appraising their validity, and drawing appropriate conclusions.

Response: We thank the reviewer for the comment. The results were presented as mean ± SEM from triplicate experiments, analyzed using ANOVA to ensure consistency. Due to the limited sample size (n=3), further inferential statistics (e.g., p-values) were not applied to avoid overinterpretation. However, conclusions were supported through complementary molecular docking, MD simulations, and MM-GBSA analyses, providing robust validation of the findings.

Discussion : The discussion section should be revised and focused on interpretation, relationships with reports from other scholars in the field, and implication of the obtained results.

Response: Dear reviewer, we have revised the discussion as per the suggestions

Reviewer #3: I recommend the acceptance of the manuscript. One additional comment I just mention that author should write the unit of binding energy in the manuscript.

Response: We thank the reviewer for the suggestion. The unit of binding energy (kcal/mol) has been added in the text and Table 4.

7. PLOS authors have the option to publish the peer review history of their article (what does this mean?). If published, this will include your full peer review and any attached files.

Do you want your identity to be public for this peer review? For information about this choice, including consent withdrawal, please see our Privacy Policy.

Reviewer #1: Yes: Khaled M. Darwish

Reviewer #2: No

Reviewer #3: Yes: Shashank Shekher Mishra

Response: It okay

---

## [Editor Report · Decision Letter 2]

22 Apr 2025

Investigation of the inhibitory potential of secondary metabolites isolated from Fernandoa adenophylla against Beta-glucuronidase via molecular docking and molecular dynamics simulation studies

PONE-D-25-04376R2

Dear Dr. Rauf,

We’re pleased to inform you that your manuscript has been judged scientifically suitable for publication and will be formally accepted for publication once it meets all outstanding technical requirements.

Kind regards,

Viet Phong Nguyen, Ph.D.

Academic Editor

PLOS ONE

---

## [Editor Report · Acceptance letter]

PONE-D-25-04376R2

PLOS ONE

Dear Dr. Rauf,

I'm pleased to inform you that your manuscript has been deemed suitable for publication in PLOS ONE. Congratulations! Your manuscript is now being handed over to our production team.

Kind regards,

on behalf of

Dr. Viet Phong Nguyen

Academic Editor

PLOS ONE